# DRG2 Deficient Mice Exhibit Impaired Motor Behaviors with Reduced Striatal Dopamine Release

**DOI:** 10.3390/ijms21010060

**Published:** 2019-12-20

**Authors:** Hye Ryeong Lim, Mai-Tram Vo, Dong Jun Kim, Unn Hwa Lee, Jong Hyuk Yoon, Hyung-Jun Kim, Jeongah Kim, Sang Ryong Kim, Jun Yeon Lee, Chae Ha Yang, Hee Young Kim, June-Seek Choi, Kijeong Kim, Esther Yang, Hyun Kim, Seongsoo Lee, Byung Ju Lee, Kyungjin Kim, Jeong Woo Park, Chang Man Ha

**Affiliations:** 1Research Division and Brain Research Core Facilities of Korea Brain Research Institute, Daegu 41068, Korea; hrsz@kbri.re.kr; 2Department of Biological Sciences, University of Ulsan, Ulsan 44610, Korea; vomai_tram@yahoo.com (M.-T.V.); kdjehdwns123@naver.com (D.J.K.); unnhwa@naver.com (U.H.L.); bjlee@ulsan.ac.kr (B.J.L.); 3Dementia Research Group and Neurodegenerative Disease Group, Korea Brain Research Institute, Daegu 41068, Korea; jhyoon@kbri.re.kr (J.H.Y.); kijang1@kbri.re.kr (H.-J.K.); 4Department of Brain and Cognitive Science, DGIST, Daegu 42988, Korea; alice860413@naver.com (J.K.); kyungjin@dgist.ac.kr (K.K.); 5School of Life Sciences, BK21 Plus KNU Creative BioResearch Group, Kyungpook National University, Daegu 41566, Korea; srk75@knu.ac.kr; 6College of Korean Medicine, Daegu Haany University, Daegu 42158, Korea; junyeon88@gmail.com (J.Y.L.); chyang@dhu.ac.kr (C.H.Y.); vet202001@gmail.com (H.Y.K.); 7Department of Psychology, Korea University, Seoul 02841, Korea; j-schoi@korea.ac.kr; 8School of Exercise and Sport Science, University of Ulsan, Ulsan 44610, Korea; kijeongk@ulsan.ac.kr; 9Department of Anatomy, College of Medicine, Korea University, Seoul 02841, Korea; esther1122@korea.ac.kr (E.Y.); kimhyun@korea.ac.kr (H.K.); 10Gwangju Center, Korea Basic Science Institute (KBSI), Gwangju 61886, Korea; soolee@kbsi.re.kr

**Keywords:** Developmentally regulated GTP-binding protein 2 (DRG2), Dopamine release, Motor deficiency, Dopaminergic neurons, Motor coordination, Striatum

## Abstract

Developmentally regulated GTP-binding protein 2 (DRG2) was first identified in the central nervous system of mice. However, the physiological function of DRG2 in the brain remains largely unknown. Here, we demonstrated that knocking out DRG2 impairs the function of dopamine neurons in mice. DRG2 was strongly expressed in the neurons of the dopaminergic system such as those in the striatum (Str), ventral tegmental area (VTA), and substantia nigra (SN), and on neuronal cell bodies in high-density regions such as the hippocampus (HIP), cerebellum, and cerebral cortex in the mouse brain. DRG2 knockout (KO) mice displayed defects in motor function in motor coordination and rotarod tests and increased anxiety. However, unexpectedly, DRG2 depletion did not affect the dopamine (DA) neuron population in the SN, Str, or VTA region or dopamine synthesis in the Str region. We further demonstrated that dopamine release was significantly diminished in the Str region of DRG2 KO mice and that treatment of DRG2 KO mice with l-3,4-dihydroxyphenylalanine (L-DOPA), a dopamine precursor, rescued the behavioral motor deficiency in DRG2 KO mice as observed with the rotarod test. This is the first report to identify DRG2 as a key regulator of dopamine release from dopamine neurons in the mouse brain.

## 1. Introduction

Dopamine is a major neurotransmitter and plays an important role in motor control, and its dysfunction in the dopamine (DA) neuron circuitry affects various neuronal disorders such as Parkinson’s disease (PD), schizophrenia, and depression [1,2,3]. Midbrain dopamine neurons are located in the substantia nigra (SN), ventral tegmental area (VTA), and retrorubral area and dopamine neurons in the SN project into the dorsal striatum (Str) via the nigrostriatal pathway, whereas dopamine neurons in the VTA project into the ventral Str, limbic systems, and prefrontal cortex. Recent studies have shown that the dopamine neurons of the midbrain are particularly vulnerable to microtubule (MT) disruptions and that MT destabilization leads to a decrease in the dopamine level in the VTA after intracerebral hemorrhage [4,5]. Dopamine neurons from the SN send long axons to the Str to control locomotor activity, and MTs participate in axonal secretory vesicle transport [6]. In the case of PD, MT destabilization causes the accumulation of dopamine vesicles in soma and may trigger dopamine oxidation and selective dopamine neuron death [7]. Thus, MT stabilization may protect dopamine neurons and improve motor dysfunction and animal behavior. The Rac1 protein, a member of the subfamily of Rho-GTPases, interacts with MTs [8], stabilizes MTs [9], and participates in the regulation of dopaminergic cell death, neuronal polarization, vesicle trafficking, and axon growth through cytoskeletal organization [10,11,12]. Developmentally regulated GTP-binding proteins (DRGs) are a novel class of GTPases [13] that consist of two closely related proteins, DRG1 and DRG2 [14]. DRG1 and DRG2 share 55% amino acid sequence identity, but interact with different molecules, DFRP1 and DFRP2, respectively [15], suggesting that they have distinct functions. However, interestingly, recent reports have suggested that both of these proteins play roles in the regulation of MT polymerization. DRG1 interacts with MTs and promotes MT polymerization [16], while DRG2 interacts with Tau and regulates MT polymerization [17]. Moreover, it has been reported that DRG2 regulates MT stability by modulating Tau phosphorylation. DRG2 interacts with Rab5 on vesicular endosomes and is involved in the regulation of endosomal Rab5 activity and endosome trafficking [18]. DRG2 depletion extends the expression of EGFR on Rab5-containing endosomes and increases the activity of Akt and inhibitory phosphorylation of GSK3β, thus reducing the phosphorylation of Tau, which leads to an enhanced interaction between Tau and MT and hyperstabilization of MTs [19]. Considering the involvement of Tau [20] and MTs [7] in neurodegenerative diseases, we predict that DRG2 may play an important role in the pathogenesis of neurodegenerative diseases. 

In this study, we investigated the expression pattern of DRG2 in the adult mouse brain and the impact of DRG2 depletion on mouse brain function using DRG2 knockout (KO) mice. DRG2 was predominantly expressed in neurons rather than glial cells, and especially colocalized with tyrosine hydroxylase (TH) neurons. We further demonstrated that DRG2 KO mice displayed defects in dopamine release from the Str region and motor function, while DRG2 depletion did not affect the dopamine neuron population or dopamine synthesis in the Str region. These results provide new insights into how DRG2 depletion leads to defects in the function of dopamine neurons.

## 2. Results

### 2.1. Growth Retardation and Skeletal Defects in DRG2 KO Mice

To determine the physiological function of DRG2, we first observed changes in the phenotype of DRG2^−/−^ (homozygous-mutant) mice compared to DRG2^+/+^ (wild-type (WT) control) and DRG2^+/−^ (heterozygous-mutant) mice. The absence of DRG2 mRNA and protein in the cerebral cortex of DRG2^−/−^ mice was confirmed by RT-PCR (Figure 1A) and western blotting (Figure 1B). DRG2^−/−^ mice were significantly smaller at birth (1 day old) than their DRG2^−/−^ and DRG2^+/−^ littermates, and this difference in size persisted into adulthood (12 weeks old) (Figure 1C). Consistently, the body weight of DRG2^−/−^ mice was significantly less than that of WT control mice (Figure 1D). To confirm whether the small body size of DRG2^−/−^ mice is associated with skeletal changes, we analyzed bone tissues from postnatal 1-day-old pups using micro-computerized tomography (CT). While the long bones in the appendicular skeleton of DRG2^+/−^ mice had apparently normal bone density, those of the DRG2^−/−^ mice were approximately 8~33% shorter than those of the WT control mice. In addition, DRG2^−/−^ mice showed significantly reduced mineralization of the skull and increases in unossified areas in the anterior fontanel (Figure 1E). We further examined postnatal survival rates to identify the growth retardation events leading to the observed physiological changes. DRG2^−/−^ mice showed significantly (*p* < 0.001) reduced survival, and the survival rate of DRG2^−/−^ mice after 1 year was only 20% of the rate of WT mice, whereas DRG2^+/−^ mice showed no significant change in survival (*p* > 0. 05).

### 2.2. Regional Distribution of DRG2 in Mouse Brain

To identify how the growth retardation induced by DRG2 deficiency affects brain function, we characterized the detailed phenotypic expression pattern of mouse DRG2. DRG2 was identified as highly expressed in the mouse embryonic brain and many human tissues such as the skeletal muscle, kidneys, liver, heart, and brain in previous reports [14,21]. However, the gene and protein expression in specific brain regions is still unknown. Thus, we first used an antibody specific for DRG2 to stain each organ tissue and identified the brain-region specificity of DRG2. The DRG2 protein was expressed strongly in the brain, lung, and spleen, whereas it was expressed weakly in the thymus, liver, testis, kidney, and lungs (Figure 2A). In the adult mouse brain, the DRG2 protein was predominantly expressed in extracts from the cortex (CTX), hippocampus (HIP), cerebellum (CB), and SN, whereas less protein was detected in the hypothalamus (HYP) and Str (Figure 2B). The protein expression levels of DRG2 were not significantly different during postnatal development among brain regions with high expression (Figure 2C). We further confirmed the expression pattern of DRG2 in the adult mouse brain using in situ hybridization and immunohistochemistry to determine the function of DRG2 in the brain. In situ hybridization revealed widespread expression of DRG2 across the mouse brain, but strong signals for the DRG2 protein including in the suprachiasmatic nucleus (SCN) were consistently detected by western blotting (Figure 2D). Immunohistochemical analysis showed that the expression pattern of the DRG2 protein was similar to that of the DRG2 transcript, but the protein expression was different at the spatial level: high expression levels were seen in the accessory olfactory bulb, CTX, HIP, SN, CB and parabrachial nucleus (PB), while moderate expression was observed in the HYP, nucleus accumbens (NAc), and thalamus (THs). As shown in the magnified image, the DRG2 protein was detected as dots in the perinuclear region and dominantly in Purkinje cell bodies in the cerebellum. The DRG2 protein was also highly expressed in layer VI of the cerebral cortex, cornu ammonis 3 (CA3) of the HIP, the substantia pas compacta, and the substantia pars reticulate (Figure 2E(a–i)). We next investigated the DRG2 protein expression pattern in the coronal section of the mouse brain. DRG2 protein expression was strong in all cerebral cortical layers in different areas such as the somatomotor and somatosensory area, retrosplenial area, parietal association areas, auditory areas, visual area, piriform area, and dentate gyrus (DG), cornu ammonis 1 (CA1), CA3 in the limbic system, and entorhinal area during hippocampal formation. DRG2 was also highly expressed in the basomedial amygdalar nucleus, thalamus, and midbrain (Figure 2E(j–l)).

To further determine the cellular specificity of DRG2 expression in the mouse brain, we conducted immunohistochemical analysis to compare the expression of DRG2 with that of the following cell-type specific markers: the astrocyte marker GFAP; microglial marker Iba1; and neuronal marker Tuj1. DRG2 highly colocalized with Tuj1-positive neurons in brain regions with strong expression such as the SN, HIP, and CTX, rather than with GFAP-positive astrocytes or Iba1-positive microglia in these brain regions (Figure 3A and Appendix A). To confirm this, we isolated neurons, astrocytes, and microglia and analyzed the expression of DRG2 in these cells using western blot. As shown in Appendix A, DRG2 expression was detected from all type of cells, but the DRG2 level in neurons was higher than those in astrocytes and microglia. We then determined whether DRG2 was differentially expressed between excitatory and inhibitory neurons. In the HIP, DRG2 mRNA was mainly expressed in the DG and Cornu Ammonis areas (CA1 and CA3) (Figure 3B,C). We performed double in situ hybridization using antisense probes for excitatory neuron markers such as vesicular glutamate transporters 1 and 2 (VGLUT1 and VGLUT2, respectively) and for inhibitory neuron markers such as GAD65 and GAD67. VGLUT1 is mainly expressed in the cerebral cortex, HIP, and cerebellum, whereas VGLUT2 is preferentially expressed in the TH and HYP [22]. Consistently, we found that DRG2 was expressed in the DG and CA1 regions of the HIP, which highly expressed VGLUT1, but VGLUT2 was not detected (Figure 3D). DRG2 colocalized with GAD65 and GAD67 even though the expression levels of GAD65 and GAD67 were very low in the HIP (Figure 3E). These findings indicated that double in situ hybridization was successful and that DRG2 colocalized with not only excitatory neurons, but also inhibitory neurons in the HIP (Figure 3D,E).

### 2.3. DRG2 Depletion Causes Motor Deficit and Increases Anxiety in Mice

To clarify the effects of DRG2 depletion on the excitatory and inhibitory neurons of brain regions with high DRG2 expression, we performed a series of behavior tests using DRG2^−/−^ mice. We first utilized the negative geotaxis test, which evaluates motor coordination and vestibular sensitivity [23]. In this test, the latency time required for mice to reorient themselves into a head-upward position on an inclined plane was recorded, and the time was recorded as 60 s for mice that failed to reorient themselves into the head-upward position. Even though an initial delay in reorientation was observed in postnatal day 4 (P4) pups of WT mice, all WT mice readily reoriented themselves gravitationally into the head-upward position within 60 s on all postnatal days tested. In contrast, pups of DRG2^−/−^ mice failed to complete this test due to their inability to stand on the inclined plane until P8, and beginning at P10, DRG2^−/−^ mice were able to complete the test. Post hoc analysis showed significant differences between the DRG2^−/−^ mice and WT control mice at P4 (*p* = 0.002), P6 (*p* = 0.002) and P8 (*p* = 0.015) (Figure 4A). We also performed a rotarod test to examine motor coordination and motor learning skills at different ages. The fixed-speed version of the rotarod task was applied to test WT and DRG2^−/−^ mice aged six weeks, 12 weeks, and 40 weeks. Compared to WT mice (300 s), female DRG2^−/−^ mice aged six weeks showed a significant reduction in motor performance (243 ± 22 s). Female DRG2^−/−^ mice aged 12 weeks or 40 weeks stayed on for much shorter times (123 ± 46 s and 86 ± 19 s, respectively) than WT mice and 6-week-old DRG2^−/−^ mice (Figure 4B). The reduced times for the DRG2^−/−^ mice, except the 40-week-old DRG2^−/−^ mice, observed in the rotarod test were improved by repeated tests one week later. These results suggest that DRG2^−/−^ mice had a defect in motor coordination, but had normal motor learning skills. We next evaluated whether DRG2 depletion affects learning and memory behaviors using Y-maze tasks. Both male and female DRG2^−/−^ young (6-week-old) and adult (12-week-old) mice showed performances in Y-maze tasks similar to those of WT mice in terms of percentage of alternation triplets, number of total entries, and total distance at 12 weeks of age (Figure 4C). These data suggest that DRG2 depletion does not affect memory performance in mice. We further evaluated anxiety- and fear-related behaviors of DRG2^−/−^ mice using an elevated plus-maze test. The data showed that both male and female 12-week-old DRG2^−/−^ mice spent significantly less time in the open arms than WT mice of the same age (Figure 4D), suggesting that DRG2 depletion increased anxiety-like behavior in mice.

### 2.4. Abnormal Behaviors of DRG2-Deficient Mice Are Caused by Dopamine Release Alterations

Dopamine neurons extend their axons into the Str and secrete dopamine, which binds to dopamine receptors D1 and D2 expressed in the Str [24]. The alteration in the number of striatal dopamine neurons and the dopamine release can affect striatal-relevant behaviors such as anxiety, motor coordination, locomotion, and sensitivity to reward. The alterations in motor coordination and anxiety behaviors in DRG2^−/−^ mice prompted us to test whether DRG2 depletion causes defects in dopamine neurons. Thus, we applied double in situ hybridization using antisense probes against DRG2 and TH to analyze the expression pattern of DRG2 in dopamine neurons. High expression levels of DRG2 were detected in the TH-positive neurons in the VTA and SN (Figure 5A). Individual mRNA transcripts appeared as bright dots within cells, and the number of DRG2 mRNA dots was significantly higher in the TH-positive cells (11.85 ± 1.26 and 13.10 ± 0.84) than in the TH-negative cells (3.2 ± 0.52 and 2.95 ± 0.64) in the SN and VTA regions, respectively (Figure 5B,C). Our data suggest that DRG2 may play roles in dopamine neurons and that DRG2 depletion may cause dopamine neuron dysfunction.

Both dopamine-producing cell death in the substantia nigra pars compacta (SNc) and decreased striatal dopamine release can lead to dopamine neuron dysfunction [25,26,27]. We thus determined whether DRG2 depletion induces alterations in the population of TH neurons or striatal dopamine release. TH neurons were stained with anti-TH antibody and detected using secondary antibody conjugated with horseradish peroxidase complex (Figure 5D,E) or Alexa Fluor 488/594 (Appendix A). The TH intensity and TH-positive neurons were quantified. Unexpectedly, there were no significant differences in the intensity and distribution of TH-positive neurons in the SN, VTA, NAc, and Str areas between 24-month-old WT and DRG2^−/−^ mice (Figure 5D,E and Appendix A). In addition, DRG2 depletion did not induce any alterations in the target area of the nigrostriatal or mesolimbic pathways of TH neurons (Appendix A). These data suggest that DRG2 depletion does not affect dopamine neuron death. We next analyzed the level of striatal monoamines, which are involved in the regulation of motor behaviors. Interestingly, the dopamine content (~29%, *p* = 0.0007), 3-methoxytyramine (3-MT) level (~27%, *p* = 0.0062) and epinephrine (EPI) level (~24%, *p* = 0.0289) in the Str were significantly reduced in DRG2^−/−^ mice. However, there were no significant differences in the levels of the dopamine metabolites 3,4-dihydroxyphenylacetic (DOPAC), serotonin (5-HT), and serotonin between WT and DRG2^−/−^ mice (Figure 5F). These data suggest that DRG2 depletion results in nigrostriatal dopaminergic dysfunction and decreased dopamine levels in the Str. We next tested whether DRG2 depletion affects presynaptic dopamine release using mouse brain slices containing the SN and Str. The brain slices were stimulated by a bipolar stimulating electrode, and dopamine release was measured using a carbon-fiber recording microelectrode. Dopamine release from the brain slices of DRG2^−/−^ mice was significantly lower than that from brain slices of WT mice (Figure 5G). To determine whether the reduced dopamine release caused altered motor coordination behavior in DRG2^−/−^ mice, we treated DRG2^−/−^ mice with L-DOPA and performed the rotarod test. Compared with no treatment or PBS treatment, the administration of L-DOPA significantly increased the motor coordination of the DRG2^−/−^ mice in the rotarod test (Figure 5H). Collectively, our results suggest that DRG2 depletion reduces the release of nigrostriatal dopamine, which leads to altered motor coordination and anxiety behaviors.

## 3. Discussion

DRG2 has been suggested to play a key role in cell cycle regulation in cell growth and differentiation control [28,29] inducing mitochondrial function and affecting cell migration via the regulation of MT dynamics and Golgi fragmentation [19,30] in overexpression or knockdown experiments. These in vitro results suggest that DRG2 may play important roles in in vivo developmental stages and behavioral differences. To provide insight into the functional roles of DRG2 in mouse development, we report here, for the first time, that DRG2-deficient mice exhibit a significant growth delay and reduced body weight accompanied by defective osteogenesis and a short life span. Interestingly, DRG2 overexpression caused a decrease in bone mass and increases in the number and activity of osteoclasts in vivo in a previous work [31]. Conversely, in micro-CT analyses of DRG2-deficient mice, it was clearly identified that unossified areas were significantly increased in the skull and that bone formation was dramatically reduced in the appendicular skeleton. The maintenance of bone mass and integrity requires a precise balance between osteoclasts and osteoblasts. However, there were no significant differences in whole-brain size, whole-brain weight, or brain structure, even though DRG2-deficient mice exhibited reduced ossification of the skull (Appendix A). These results suggest that DRG2 may play a critical role in bone remodeling by regulating bone formation and bone resorption during different developmental stages, but does not affect embryonic or postnatal brain development. DRG2 was robustly expressed in the brain such as in the SN, CTX, CB, and HIP, which are areas related to motor behavior and learning and memory. DRG2 was predominantly expressed in neuronal cells including TH neurons, glutamatergic neurons, and GABAergic neurons rather than glial cells in these brain regions. These findings suggest that DRG2 may control motor behavior or other behaviors during developmental stages. DRG2-deficient mice performed worse than WT control mice in the rotarod test, negative geotaxis test, and elevated plus-maze test. The WT control and DRG2-deficiency mice used for these behavior tests were the same age. However, DRG2-deficient mice showed lower body weight than the WT control mice. It is possible that the lower body weight of DRG2-deficient mice may lead to poor behavioral performance. However, it has been reported that lower body weight did not affect [32] or even increased [33] the rotarod performance of mice. Thus, it is unlikely that the reduced body weight of DRG2-deficient mice led to poor performance of those mice in the rotarod test. The Y-maze test showed no significant difference in spatial working memory. 

The brain dopamine system is organized into four distinct pathways: the nigrostriatal, mesolimbic, mesocortical, and tuberoinfundibular pathways. The nigrostriatal pathway transmits dopamine from the SNc to the dorsal Str, which includes the caudate putamen [34], and the mesolimbic pathway transmits from the VTA to the ventral Str, which includes both the NAc and the olfactory tubercle [35]. Both the dorsal and ventral Str are primarily composed of medium spiny neurons (MSNs), and MSNs contain different dopamine receptors [36,37]. The different dopamine receptor subtypes in MSNs are each associated with particular neural functions such as spatial working memory, cognition, and motor function [38,39,40]. 

In our results, DRG2 was highly expressed in the nigrostriatal and mesolimbic pathways of brain regions such as the SN, Str, VTA, NAc, AOB, and CTX, and spatial working memory and the TH neuron level in the NAc were not changed by DRG2 deficiency. Thus, the regional expression pattern in the brain and a series of behavioral test results indicate that DRG2 may play more important roles in motor coordination than in working memory by dopamine signaling to dopamine neurons. Additionally, knocking down DRG2 expression significantly reduced dopamine, 3-MT, and epinephrine levels in the Str, whereas DOPAC, homovanillic acid (HVA), and 5-HT levels were not significantly different. Epinephrine and norepinephrine are produced from dopamine via catecholaminergic pathways and cause various behavioral effects including motor deficits [41,42]. 3-MT is a metabolite of dopamine formed by the enzyme catechol-O-methyl transferase (COMT), and DOPAC and HVA are produced by the enzyme monoamine oxidase (MAO). These data suggest the possibility that DRG2 may regulate MAO enzymatic activity, and further studies are required to elucidate this mechanism. 5-HT modulates mood control and the pleasure response in the ventral Str, whereas 5-HT in the dorsal Str controls motor behavior [43]. However, our results indicated that the 5-HT level was not significantly changed in the DRG2-deficient Str, indicating that DRG2 affected motor behaviors via the dopamine pathway. According to our results, the increased dopamine level and normal dopamine synthesis in the Str and recovered motor function in the rotarod test induced by L-DOPA treatment strongly suggest that DRG2 regulates dopamine release in striatal dopamine neurons, which controls dopamine-dependent behaviors such as motor coordination and anxiety. It is well-known that dopamine signaling is associated with working memory [44]. It is not clear how low the level of dopamine in DRG2-deficient mice decreases motor coordination without affecting working memory. It is possible that the low level of dopamine observed in DRG2-deficient mice may be enough to activate the dopamine receptor subtype involved in the working memory, or DRG2 deficiency may increase the expression level of the working memory-related dopamine receptor. Further studies are required to clarify these. Interestingly, DRG2 is one of the genes on chromosome 17 at 17p11.2, which is related to Potocki–Lupski syndrome (PTLS) [45] and Smith–Magenis syndrome (SMS) [46] by point duplication and deletion, respectively. SMS patients exhibit craniofacial abnormalities, developmental delays, anxiety, hyperactivity, and circadian abnormalities [47,48]. Our preliminary analyses revealed that DRG2-deficient mice had hyperactivity and circadian abnormalities (H.R.L. and C.M.H., unpublished data). These symptoms are remarkably similar to DRG2-deficient mouse behavior. Accordingly, DRG2-deficient mice may provide an animal model to study dopamine-dependent developmental disorders. Further studies may help explain the pathological effects of DGR2 in these disorders. 

In summary, this study offers new insight into the role of DRG2 in mouse development and represents the first demonstration of DRG2 functions in dopamine-mediated behaviors. DRG2-deficient mice exhibited poor motor coordination and motor behavior deficits in the rotarod test, negative geotaxis, and anxiety test, whereas there were no differences in motor skill learning and memory in the Y-maze test. These motor behavioral deficits were caused by striatal dopamine release without the loss of dopamine neurons, and the behavioral abnormalities were rescued by L-DOPA treatment. Our findings suggest that DRG2 is a key regulator of dopamine release from dopamine neurons and may be a potential therapeutic target in disorders involving motor dysfunction.

## 4. Materials and Methods

### 4.1. Mouse Husbandry

DRG2^+/−^-C57BL/6 (heterozygous-mutant) and DRG2^−/−^-C57BL/6 (homozygous-mutant) mice and the genotyping primer set were described previously [18]. DRG2^+/+^-C57BL/6 (wild-type) and DRG2^−/−^-C57BL/6 (homozygous-mutant) mice were identified by genotyping after heterozygous-mutant mice were bred, and DRG2^−/−^ mice were backcrossed onto the C57BL/6 WT background for 10 generations and then maintained at the same animal facility at the Korea Brain Research Institute (KBRI). Mice were housed in groups of 2–5 animals per cage with ad libitum access to standard chow and water in 12/12 light/dark cycle with “lights-on” at 07:00, at an ambient temperature of 20–22 °C and humidity (about 55%) through constant air flow. The well-being of the animals was monitored on a regular basis. All of the experimental design was reviewed and approved (2018. 08. 07) by the Institutional Animal Care Use Committee (IACUC) of the KBRI (IACUC-18-00018). The behavior tests were conducted during the light phase of the circadian in the testing room. The rotarod test, Y-maze test, and elevated plus-maze test were performed using the same mice. On testing day, mice were habituated to the testing room and given 30 min to acclimate in their homecage.

### 4.2. Rotarod Test

A rotarod test was performed to assess motor skill learning. Before the training sessions, the mice were habituated to stay on the stationary drum for 3 min. Habituation was repeated every day for 1 min just before the session. Mice were tested on a rotating rod (Yusung Lab, Seoul, Korea) at a fixed speed of 20 rpm for 300 s, and mouse performances were evaluated over three trials per session on three consecutive days. A resting time of 180 s was allowed between each trial. The end of a trial was defined as the time when the mouse fell off the rod or when the mouse reached 300 s. The latency time to fall was recorded for each trial.

### 4.3. Y-Maze Test

For spatial learning, we used a homemade Y maze that included three identical arms (50-cm long, 16-cm wide, and 32-cm high, set at 120° to each other) illuminated by a dim light. Visual details in the testing room were kept constant across training sessions. The day prior to the start of the test, mice were habituated by being allowed to freely explore the maze for 5 min. Experimental animals were located at the start point and allowed to visit the three open arms for 8 min. During intertrial intervals, each experimental animal was housed in its home cage. For quantitative analysis, the percentage of alternation triplets and total number of arm entries were recorded online with a system for video analysis of movements (Ethovision 2.3, Noldus, Ltd., Wageningen, NL, USA).

### 4.4. Elevated Plus-Maze Test

An elevated plus-maze test was used to measure anxiety-like behavior. The apparatus consisted of a plus-shaped maze with two arms enclosed by sidewalls (50.8-cm long, 12-cm wide, and 40.6-cm high) and two open arms (50.8-cm long and 12-cm wide) that when extended formed a common central platform (10 × 10 cm) raised 72.4 cm above the floor. Mice were habituated by placing them in the center of the maze and allowing to explore for 10 min. After that, each mouse was individually placed in the central platform facing the open arm and allowed to explore the maze freely for 5 min. After each session, the apparatus was cleaned with 10% ethanol. Mouse activity was recorded online with a system for video analysis of movements (Ethovision 2.3, Noldus, Ltd., Wageningen, NL, USA). The percentage of time spent in the open arms was used as an index of anxiety-like behavior.

### 4.5. Negative Geotaxis Test

Negative geotaxis was evaluated by following the protocol of a previous report [23]. Briefly, mice were habituated by placing them on the grid for 5 min and were tested by placing each neonate (P4 or older) on an inclined rough surface grid at an approximately 35° with the mouse head facing downward. The pups were scored as follows: able to turn around and climb upward within 60 s, tried but failed to turn around within 60 s, or failed to perform the test due to a clear inability to stand as a result of significant muscle weakness and almost complete paralysis. 

### 4.6. In Situ Hybridization

In brief, frozen sections (14 µm) were cut coronally through the hippocampal formation. The sections were thaw-mounted onto Superfrost Plus Microscope Slides (#12-550-15, ThermoFisher Scientific, Waltham, MA, USA). The sections were fixed in 4% paraformaldehyde for 10 min, dehydrated in increasing concentrations of ethanol for 5 min, and then air dried. The tissue sections were then pretreated for protease digestion for 10 min at room temperature. Probe hybridization and amplification were performed at 40 °C using a HybEZ hybridization oven (Advanced Cell Diagnostics, Hayward, CA, USA). The probes used in this study were three synthetic oligonucleotides complementary to nucleotides (nt) 2–306 of Mm-DRG2, nt 62–3113 of Mm-Gad1-C2, nt 552–1506 of Mm-Gad2-C3, nt 464–1415 of Mm-Slc17a7/Vglut1-C3, nt 1986–2998 of Mm-Slc17a6/Vglut2-C2, and nt 483–1603 of Mm-TH-C2 (Advanced Cell Diagnostics, Hayward, CA, USA). The labeled probes were conjugated to Alexa Fluor 488, Atto 550, and Atto 647. The sections were hybridized with the labeled probe mixture at 40 °C for 2 h per slide. Unbound hybridization probes were removed by washing the sections three times with a 1x wash buffer at room temperature for 2 min. The steps for signal amplification included incubations at 40 °C with Amplifier 1-FL for 30 min, Amplifier 2-FL for 15 min, Amplifier 3-FL for 30 min, and Amplifier 4 Alt B-FL for 15 min. Each amplifier solution was removed by washing with the 1x wash buffer at room temperature for 2 min. The slides were viewed, analyzed, and photographed using TCS SP8 Dichroic/CS (Leica, Wetzlar, Germany), and the ImageJ program (NIH) was used to analyze the images. 

### 4.7. Immunoblotting and Immunostaining and Quantification of TH Positive Neurons 

To collect mouse tissues, mice were deeply anesthetized with a 2.5% solution of Avertin and transcardially perfused with ice-cold PBS solution to remove the blood. Each organ tissue was quickly dissected and snap frozen in liquid nitrogen before being stored at −80 °C. Mouse tissue lysates were homogenized with RIPA buffer (Sigma-Aldrich, St. Louis, MO, USA) and T-PER buffer (ThermoFisher Scientific, Waltham, MA, USA) containing protease and phosphatase inhibitors. The tissue extracts were centrifuged at 14,000 rpm at 4 °C for 20 min twice before collecting the supernatant. Equal amounts of protein were separated by SDS-PAGE and transferred to PVDF membranes (Bio-Rad, Hercules, CA, USA). The membranes were then blocked with 5% (*w*/*v*) nonfat milk (Sigma-Aldrich, St. Louis, MO, USA) in TBS for 1 h at room temperature. Anti-DRG2 and anti-actin primary antibodies (anti mouse-DRG2 (Proteintech, Rosemont, IL, USA) and anti–β-actin (Cell Signaling, Danvers, MA, USA)) were diluted in 5% normal goat serum in TBS plus 0.05% Tween 20 (TBS) and incubated with the membranes at 4 °C overnight. Unbound antibodies were removed by washing the membranes in TBST four times for 15 min each time at room temperature. Secondary antibodies conjugated with HRP were incubated for 1 h at room temperature. After extensive washing with TBST, immunoreactivity was detected using a GE HealthCare Chemi image documentation system. Immunocytochemistry was performed on samples taken from mice deeply anesthetized with a 2.5% Avertin solution and transcardially perfused with a PBS solution followed by 4% paraformaldehyde (PFA) in PBS. Dissected brains were kept in 4% PFA in PBS overnight at 4 °C, and the postfixed brains were kept in 4% PFA in a 10% sucrose 1x PBS buffer for at least overnight at 4 °C. The fixed brains were then embedded in an O.C.T. compound (Sakura Finetek, Torrance, CA, USA) before being sectioned at 50–100 μm thickness by using a cryostat (CM1850, Leica, Wetzlar, Germany). Each brain section was washed once with PBS and then permeabilized with 0.2% Triton X-100 in PBS for 1 h at room temperature, followed by blocking in 2.5% normal goat serum in PBS for 1 h at room temperature. Primary antibodies (rabbit anti-DRG2 (NOVUS, Centennial, CO, USA), mouse anti-Tuj1 (Abcam, Cambridge, UK), mouse anti-GFAP (Abcam, Cambridge, UK), goat anti-Iba1 (Abcam, Cambridge, UK), and mouse anti-TH (Sigma-Aldrich, St. Louis, MO, USA) were diluted in a blocking buffer and applied to the sections overnight at 4 °C. The sections were then washed four times with 0.1% Tween 20 in PBS for 15 min each time and stained with secondary antibodies conjugated with Alexa Fluor 488/594 (ThermoFisher Scientific, Waltham, MA, USA) for 2 h at room temperature or overnight at 4 °C, followed by DAPI staining in PBS for 15 min. A biotinylated primary antibody was used for TH neuron staining, and the addition of an anti-biotin secondary antibody was followed by incubation with a streptavidin-horseradish peroxidase complex (ABC elite kit, Vector Laboratories, Burlingame, CA, USA) for 1 h and subsequent exposure to diaminobenzidine (DAB kit, Vector Laboratories, Burlingame, CA, USA). The sections were further washed three times with 0.1% Tween 20 in PBS and two times with water. The sections were mounted with VECTASHIELD (Vector Laboratories, Burlingame, CA, USA), and images were captured by using a Nikon A1rsi/Ni-E inverted confocal microscope with a 60×, 1.4 NA oil-immersion objective lens at the Advanced Neural Imaging Center at the KBRI. The number of TH positive cells were counted manually, blinded of genotype. Beginning from the first slide of the SN, VTA, NAc, and Str section when TH positive neurons were visible, all TH neurons were counted on every six slide through the entire SN and VTA in both brain hemispheres. The estimated total number of TH positive neurons in the DRG2^−/−^ mice brain was calculated by adding the TH positive cell counts and presented as the % of TH positive neurons in the DRG2^+/+^ mice brain. TH staining intensity of Str and NAc were measured by ImageJ software. Three mice from each group were analyzed at 24 months of age.

### 4.8. Dopamine Release Measurement with High-Performance Liquid Chromatography (HPLC)

Dissected Str tissue was frozen in liquid nitrogen and stored at −80 °C. Str tissue samples were homogenized in 0.3 N perchloric acid, and then sonication was performed seven times at 12 V. The lysates were centrifuged at 18,000× *g* for 10 min at 4 °C. A sample of the supernatant (10 µL) was injected into an HPLC-ECD system consisting of an UltiMate™ 3000 ECD-3000RS (ThermoFisher Scientific, Waltham, MA, USA). Chromatographic separation was achieved on a Hypersil™ BDS C18 column (3 μm, 3 × 150 mm) including a Hypersil™ BDS guard column (28103-013001, ThermoFisher Scientific, Waltham, MA, USA) and a UniGuard™ guard cartridge holder (852-00, ThermoFisher Scientific, Waltham, MA, USA). The mobile phase was the Dionex™ mobile phase (70-3829, ThermoFisher Scientific, Waltham, MA, USA). The column temperature was 35 °C, and the flow rate was 0.5 mL/min. The ECD cell potentials were 250 mV and 400 mV. Each neurotransmitter’s peaks were identified by their retention times compared to the times of standard dopamine peaks. Concentrations were calculated by comparing the peak area of the sample chromatogram with that of standard dopamine chromatograms. Dionex™ Chromeleon™ CDS software (ThermoFisher Scientific, Waltham, MA, USA) was used for data acquisition and processing. All standard neurotransmitters including dopamine (H8502), DOPAC (850217), 3-MT (M4251), HVA (H1252), EPI (E4375), and 5-HT (H9523) were purchased from Sigma-Aldrich.

### 4.9. Measurement of Dopamine Release from the Str by Slice Fast-Scan Cyclic Voltammetry (FSCV)

Electrically evoked dopamine release in the Str was measured by FSCV. Mice were anesthetized using isoflurane and sacrificed. The brain was rapidly removed, and preparation of brain tissue began with vibratome sectioning in ice-cold, preoxygenated (95% O_2_/5% CO_2_, ≥15 min before usage) artificial cerebrospinal fluid (aCSF) containing 119 mM NaCl, 26.2 mM NaHCO_3_, 11 mM glucose, 2.5 mM KCl, 2.5 mM CaCl_2_, and 1.3 mM MgSO_4_ in 1 mM phosphate buffer at pH 7.2–7.4. Coronal brain sections (300 µm) were allowed to stand in the oxygenated aCSF at room temperature for at least 1 h. For recordings, a brain slice was transferred to a locally constructed submersion recording chamber, which was perfused at 1 mL/min with oxygenated aCSF at 32 °C. Carbon-fiber microelectrodes were prepared as previously described [49], and reference electrodes were made from chloridized silver wire. For dopamine recording in the Str, the electrode potential was linearly scanned from −0.4 to 1.2 V and back to −0.4 V versus Ag/AgCl at 400 V/s, and this scanning was repeated every 100 ms. Dopamine release was evoked every 5 min by one-pulse stimulation (monophasic, 350 µA, 4-ms pulse width) from an adjacent bipolar stimulating electrode (Plastics One, Roanoke, VA, USA) placed on the surface of the slice 100–200 µm away from the carbon-fiber microelectrode. The carbon-fiber microelectrode was positioned 100 µm below the surface of the slice. Cyclic voltammograms were recorded at the carbon-fiber microelectrode every 100 ms by means of a ChemClamp voltage clamp amplifier (Dagan Corporation, Minneapolis, MN, USA). Voltammetry recordings were acquired and analyzed using LabVIEW-based (National Instruments, Austin, TX, USA) customized software (Demon Voltammetry).

### 4.10. L-DOPA Administration

Mice were given a single intraperitoneal injection of L-DOPA (50 mg/kg; Sigma-Aldrich, St. Louis, MO, USA) (25 mg/kg) dissolved in PBS containing 0.25% ascorbic acid. Three hours after the administration of L-DOPA, the mice were tested on the rotating rod as described above.

### 4.11. Statistical Analysis

Analysis and quantification of data were performed with SigmaPlot 13.0 (Systat Software, Point Richmond, CA, USA), and data are presented as the mean ± SEM. The results were analyzed according to group or time by using one-way ANOVA, and differences were considered significant when *p* < 0.05. Significance is indicated as follows: * *p* < 0.05; ** *p* < 0.01; *** *p* < 0.001; and N.S., not significant.

## Figures and Tables

**Figure 1 ijms-21-00060-f001:**
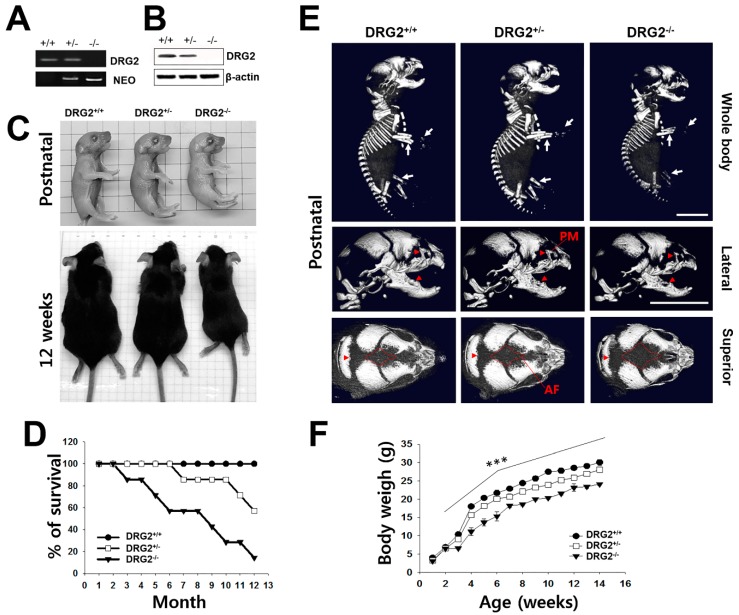
Multiple phenotypic characterization of DRG2 knock out (KO) mice. (**A**,**B**) Confirmation of DRG2 deficiency. Expression levels of DRG2 in the cerebral cortex was determined by (**A**) RT-PCR and (**B**) western blot analysis. NEO, the primer set of PGK-neo cassette targeting vector for the DRG2 KO mice. (**C**) Comparison of body size of DRG2^−/−^ mice with DRG2^+/+^, and DRG2^+/−^ littermates at postnatal day 1 (top) and 12 weeks (bottom). (**D**) Survival curve for DRG2^+/+^, DRG2^+/−^, and DRG2^−/−^ mice. n = 12 per each group. (**E**) Micro-CT scan images of DRG2^+/+^, DRG2^+/−^, and DRG2^−/−^ at postnatal day 1. Top, whole body; middle, lateral views of mice skulls; bottom, superior views of mice skulls. Arrowheads indicate reduced mineralization and arrows show reduction in the radius and ulna of the forelimb and hindlimb in DRG2^−/−^ mice. DRG2^−/−^ mice showed increased anterior fontanel (AF) and reduced premaxillary-maxillary (PM). Scale bar, 500 μm. (**F**) Body weight of DRG2^+/+^, DRG2^+/−^, and DRG2^−/−^ mice. n = 12 per each group. One-way ANOVA: *** *p* < 0.001. Error bars indicate SEM.

**Figure 2 ijms-21-00060-f002:**
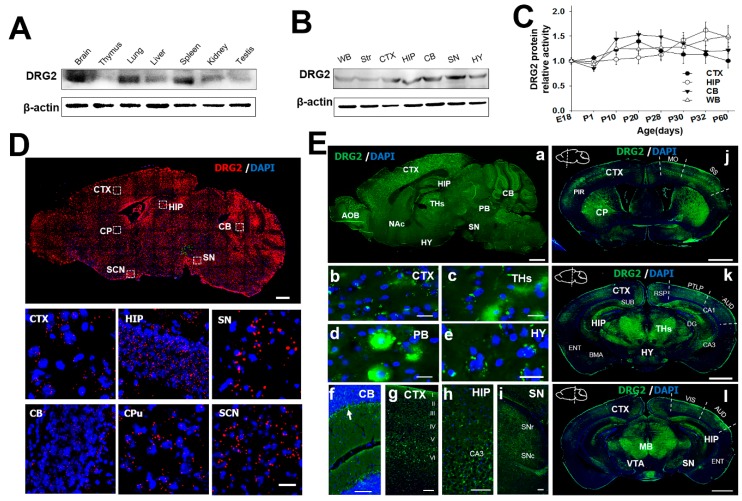
**Expression profiling of DRG2 in the mouse brain**. (**A**) Western blot analysis for DRG2 protein tissue distribution. (**B**) Western blot analysis for DRG2 expression in mouse brain regions. (**C**) Changes in DRG2 protein expression with age in the four brain regions such as CTX, cerebral cortex; HIP, hippocampus; CB, cerebellum and WB, whole brain. DRG2 expression was normalized to actin. Each data point represents the mean ±SEM (n = 6). (**D**) In situ hybridization analysis of DRG2 expression in the mouse brain. Sagittal sections of adult mouse brain were hybridized with DRG2 antisense probes. Representative images for DRG2 mRNA (red dots) and nuclei counterstained with DAPI (blue). The boxed regions were viewed at higher magnification in the bottom panels. Scale bar 1 mm. (**E**) Immunohistochemical analysis of DRG2 expression in the sagittal and coronal sections of mouse adult brain using anti-DRG2 primary antibody and Alexa-488-labeled secondary antibodies. Nuclei were stained with DAPI. (**a**) Sagittal sections of mouse brain. (**b**–**i**) Higher magnification of specific regions within sagittal image in (**a**): (**b**) CTX, (**c**) THs, thalamus (**d**) PB, parabrachial nucleus (**e**) HY, hypothalamus (**f**) CB, (**g**) CTX, (**h**) HIP, and (**i**) SN. Arrow in (**f**) indicates the purkinje cell layer. Scale bar, 100 μm. (**j**–**l**) Coronal sections of mouse brain at three different planes. Sagittal diagrams of the brain showing the plane of section were indicated within each images. AOB, accessory olfactory bulb; AUD, auditory areas; BMA, basomedial amygdala nuclear; CA1, cornu ammonis1; CA3, Cornu Ammonis3; DG, dentate gyrus; CB, cerebellum; CP, caudate putamen; CTX, cerebral cortex; ENT, Entorhinal area; HIP, hippocampus; HY, hypothalamus; MB, mid brain; Mo, somatomotor area; NAc, nucleus accumbens; PB, parabrachial nucleus; Pir, piriform cortex; PTLP, posterior parietal association areas; RSP, retrosplenial area; SCN, suprachiasmatic nucleus; SN, substantia nigra; SNc, substantia nigra pars compacta; SNr, substantia pars reticulate; SS, somatosensory area; Str, striatum; SUB, subiculum; THs, thalamus; VIS, visual area; VTA, ventral tegmental area; WB, whole brain.

**Figure 3 ijms-21-00060-f003:**
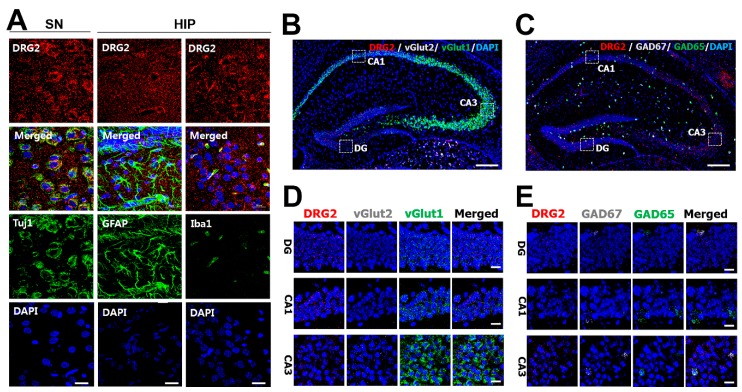
**The expression profiling of DRG2 in the mouse neurons**. (**A**) Immunohistochemical staining of neuronal marker Tuj1, astrocytes marker GFAP, microglia marker Iba1, and DRG2 in neuron cell bodies of substantia nigra (SN) and dentate gyrus (DG) of the hippocampus (HIP). Nuclei were stained with DAPI. Scale bar, 20 μm. (**B**–**E**) In situ hybridization analysis of DRG2 expression in mouse hippocampus. (**B**,**C**) Hippocampal sections of mouse brain were hybridized with antisense probes against vGlut2, GAD67 (gray dot), vGlut1, GAD65 (green dot), and DRG2 (red dot). Nuclei were stained with DAPI. Scale bar, 200 μm. The boxed regions in images of (**B**,**C**) were viewed at a higher magnification in the bottom panels (**D**,**E**), respectively. CA1 and CA3, Cornu Ammonis 1 and 3. Scale bar 20 μm.

**Figure 4 ijms-21-00060-f004:**
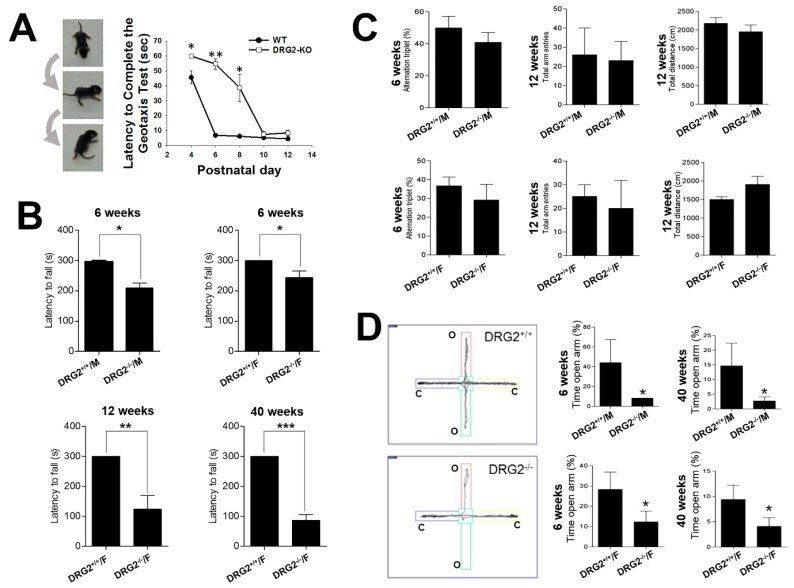
**Behavioral impairments in DRG2 deficient mice**. (**A**) Geotaxis test. (**Left**) series of photographs illustrating a mouse performing the geotaxis test. (**Right**) Graphs based on the latency to complete the geotaxis test in seconds (mean ± SEM, maximum 60 s, n = 8 per each group). * *p* < 0.05, ** *p* < 0.005. (**B**) Rotarod test. DRG2^+/+^ and DRG2^−/−^ and mice were compared at six, 12, and 40 weeks of age. At six weeks, both female and male mice were compared. At 12 and 40 weeks, female mice were compared. Data represent latency to fall (means ± SEM, maximum 300 s, n = 8 per each group). * *p* < 0.05, ** *p* < 0.005, *** *p* < 0.001. Student’s *t*-test. (**C**) Y-maze test. Both female and male of DRG2^+/+^ and DRG2^−/−^ mice were compared at six and 12 weeks of age. (**Left**) Percentage of alternation triplet; middle, total number of arm entries; (**right**) total distance. Data represent means ± SEM (n = 8 each group). (**D**) Elevated plus maze test. Both female and male of DRG2^+/+^ and DRG2^−/−^ mice were compared at 6 and 40 weeks of age. Left, representative images showing typical examples of DRG2^+/+^ and DRG2^−/−^ mice exploring in the elevated plus maze apparatus. (**Middle** and **right**), graphs based on the percentage of time spent in the open arms. Data represent means ± SEM (n = 8 per each group). * *p* < 0.05.

**Figure 5 ijms-21-00060-f005:**
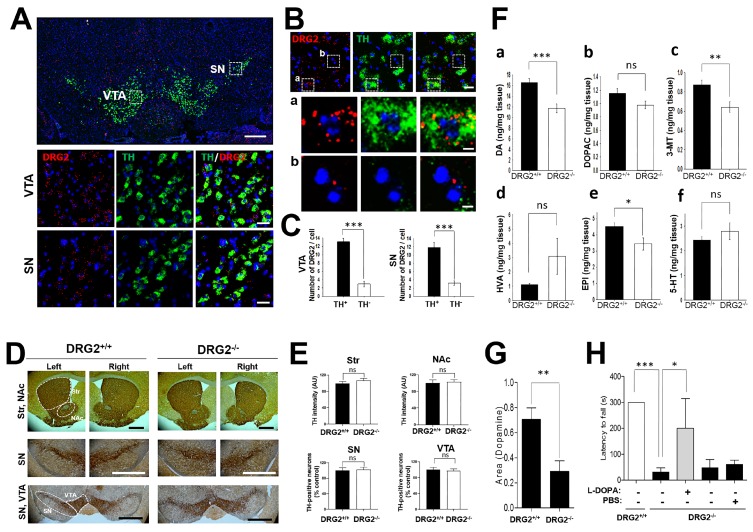
**DRG2 deficient mice show decreased release of nigrostriatal dopamine.** (**A**) In situ hybridization analysis of DRG2 and tyrosine hydroxylase (TH) expression in mouse mid brain. Coronal sections of adult mouse brain were hybridized with antisense probes against DRG2 (red) and TH (green). Nuclei were stained with DAPI. (**Top**) representative images for DRG2 and TH. Scale bar, 500 μm. (**Middle** and **bottom**) The boxed regions in the ventral tegmental area (VTA) and substantia nigra (SN) of the top panel were viewed at higher magnification in the middle and bottom panels, respectively. Scale bars, 20 μm. (**B**) Expression of DRG2 in TH-positive and -negative cells in the VTA region. (**Top**) In situ hybridization analysis of DRG2 and TH expression in the mouse VTA region. Scale bars, 20 μm. (**Middle** and **bottom**) The boxed regions in (**a**) TH-positive and (**b**) TH-negative cells of the top panel were viewed at higher magnification in the (**a**) middle and (**b**) bottom panels, respectively. Scale bars, 5 μm. (**C**) Numbers of DRG2 mRNA dots on the TH-positive or -negative cells in the SN and VTA region of (**B**) were quantified. Data represent the means ± SEM (n = 8 per each group). *** *p* < 0.0001. (**D**) Immunohistochemical staining for the TH neuron in Str, SN, NAc and VTA of 1-year old DRG2^+/+^ and DRG2^−/−^ mice. The dot area indicates the quantification regions. Scale bars, 500 μm. (**E**) Quantification of TH neurons in the images of (**D**). Graphs data represents relative TH staining intensity in the Str and NAc and number of TH positive neuron in the SN and VTA. Values obtained from DRG2^+/+^ mice were set to 100. Student’s *t*-test; ns, not significant. (**F**) HPLC analysis for striatal (**a**) dopamine, (**b**) 3,4-dihydroxyphenylacetic (DOPAC), (**c**) 3-methoxytyramine (3-MT), (**d**) homovanillic acid (HVA), (**e**) epinephrine (EPI), and (**f**) serotonin (5-HT) levels of DRG2^+/+^ and DRG2^−/−^ mice. Data represent means ± SEM (n = 8 per each group). ** *p* < 0.005, *** *p* < 0.001. (**G**) Fast-scan cyclic voltammetry (FSCV) analysis for dopamine release in the striatal slice. In the coronal brain slices, dopamine release was evoked by a single electrical stimulus pulse, and the extracellular dopamine concentration in the striatum was measured using FSCV. Data represent the area under the dopamine concentration curve (means ± SEM, n = 3 per each group). ** *p* < 0.001. (**H**) Effect of L-DOPA administration on the motor behavior of DRG2^−/−^ mice. DRG2^−/−^ mice were i.p. injected with L-DOPA (50 mg/kg) and tested on a rotating rod. Data represent the latency time to fall in the rotarod test (means ± SEM, maximum 300 s, n = 6 per each group). * *p* < 0.05, *** *p* < 0.001.

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
