# Peer review of "DRG2 Deficient Mice Exhibit Impaired Motor Behaviors with Reduced Striatal Dopamine Release"

_ijms, 2019, doi:10.3390/ijms21010060_

Round 1

Reviewer 1 Report

This article reports valuable results using DRG2 defficient mice, and provides new insights of the function of DRG2. Following points should be pointed out:

1.The method for animal care and maintainance should be described, in the view of animal wellfare.

2.Similarly, the details of animal behavioral test procedures should be described. Where these tests were pefromed? Were there periods for habituation? Did the author use same animals for those tests? These points will be valuable for the readers to reproduce the results.

3. In the Fig.3, it seems that DRG2 are colocalized with some astrocytes and microglia. I recommend some additional experiments that separate neurons and glias, such as MACS or flow cytometry.

4. In my understanding, dopamine signaling is associated with spatial working memory. The reason why there was no differences in the Y-maze test should be discussed.

5. The authors reported that the body weight of DRG2 KO mouse is significantly reduced. Does the difference of body size affect the results of geotaxis test and rotarod test? Some explanations would be needed, or should be mentioned as a limitation.

6. In the future study, I hope that studies using conditional knockout of DRG2 are performed.

Author Response

Response to Reviewers' 1 comments

Reviewer #1

Comment 1: The method for animal care and maintenance should be described, in the view of animal welfare.

Response 1: We thank the reviewer for raising this point. We describe the details of the animal welfare and added this information in the “Materials and Methods” section as bellows:

“Materials and Methods”, page 11, line 9-14: “Mice were housed in groups of 2-5 animals per cage with ad libitum access to standard chow and water in 12/12 light/dark cycle with “lights-on” at 07:00, at ambient temperature of 20–22 °C and humidity (about 55%) through constant air flow, 2-5 animals per cage with. The well‐being of the animals was monitored on regular basis. All experimental design was reviewed and approved by the Institutional Animal Care Use Committee (IACUC) of the KBRI (IACUC-2018-0020).”

Comment 2: Similarly, the details of animal behavioral test procedures should be described.

Q1) Where these tests were performed?

Q2) Where there periods for habituation?

Q3) Did the author use same animals for those tests?

These points will be valuable for the readers to reproduce the results.

Response 2:

Q1) The behavior tests were conducted during the light phase of the circadian in the testing room. On testing day, mice were brought to the testing room and given 30 min to acclimate in their homecage. We added this information to the “Materials and Methods” section (Page 11, line 14-17).

Q2) Place and periods of habituations for each behavior test were described in “Materials and Methods” section.

 Page 11, line 22-24: “Before the training sessions, the mice were habituated to stay on the stationary drum for 3 min. Habituation was repeated every day for 1min just before the session.”

 Page 11, line 32-33: “The day prior to the start of the test, mice were habituated by allowing to freely explore the maze for 5 min”

Page 11, line 42- Page 12, line 1: “Mice were habituated by placing them in the center of the maze and allowing to explore for 10 min. After then,”

page 12, line 8: “mice were habituated by placing them on the grid for 5 min and”

Q3) We used same mice for rotarod test, Y-maze test, and elevated plus-maze test. We added this information to the “Materials and Methods” section.

Page 11, line 17-18: “Rotarod test, Y-maze test, and elevated plus-maze test were performed using same mice.”

Comment 3: In the Fig.3, it seems that DRG2 are colocalized with some astrocytes and microglia. I recommend some additional experiments that separate neurons and glias, such as MACS or flow cytometry.

Response 3: We appreciate the reviewer's for this insightful comment. The Fluorescence-activated cell sorting (FACS) is one of best experiment for confirm the DRG2 expression in neuron, astrocytes and microglia. However, this experiment cannot finished within revision due day (10 days) because we cannot got the several experimental reagents for FACS. We hope to reviewer to understand that regional limitation to obtain the materials and revision period.

   Instead of a FACS experiment, we performed the immunoblot for the purified neuron, astrocytes and microglia from primary cultured cortical cells. Although primary cultured cell is developmental cell, DRG2 expression levels are not significantly changed after E18 to postnatal stages in the brain in Figure 2C result.

As you indicated, DRG2 expression could be detected in astrocytes and microglia. We changed manuscript in results, added western blot data in the supplemental figure 1E and supplemental materials and methods.

Results, page 5, line 15: from “DRG2 predominantly colocalized with Tuj1-positive neurons…… brain regions (Fig. 3A and Supplemental Fig. 1A and B)” to “DRG2 highly colocalized with Tuj1-positive neurons… brain regions (Fig. 3A and Supplemental Fig. 1A-C)”

Supplemental Figure legends, page 1, line 13-14: “(C) Western blot analysis for DRG2 in neurons, astrocytes, and microglia isolated from mouse embryo brain as described in “Materials and Methods”.”

Supplemental material and methods, page 2, line 2: “Isolation of neurons, astrocytes, and microglia”

Comment 4: In my understanding, dopamine signaling is associated with spatial working memory. The reason why there was no differences in the Y-maze test should be discussed.

Response 4: We appreciate this reviewer’s comment. We now added explanation in the “Discussion” section regarding why there was no differences in spatial working memory as follows: “It is well-known that dopamine signaling is associated with working memory (Yoon et al., 2008). It is not clear how low level of dopamine in DRG2-deficient mice decreases motor coordination without affecting working memory. It is possible that low level of dopamine observed in DRG2-deficient mice may be enough to activate dopamine receptor subtype involved in the working memory or DRG2 deficiency may increase the expression level of working memory-related dopamine receptor. Further studies are required to clarify these.”

Comment 5: The authors reported that the body weight of DRG2 KO mouse is significantly reduced. Does the difference of body size affect the results of geotaxis test and rotarod test? Some explanations would be needed, or should be mentioned as a limitation.

Response 5: We thank the reviewer for raising this point. According to your comment, we mentioned the possibility that the difference in body weight between WT control and DRG2-deficient mice may be a limitation in behavior tests and also added some explanations in the “Discussion” section as follows: “The WT control and DRG2-deficiency mice used for these behavior tests were the same age. However, DRG2-deficient mice showed lower body weight than WT control mice. It is possible that the lower body weight of DRG2-deficient mice may lead to poor behavioral performance. However, it has been reported that lower body weight did not affect (Giraldo et al., 2018) or even increased (Guerreiro et al., 2017) the rotarod performance of mice. Thus, it is unlikely that reduced body weight of DRG2-deficient mice led to poor performance of those mice in rotarod test.”

Comment 6: In the future study, I hope that studies using conditional knockout of DRG2 are performed.

Response 6: We thank and agree with the reviewer's very important comment. Now we are preparing DRG2-floxed mice and dopaminergic-specific DRG2 KO mice. With these mice, we will perform behavioral test to confirm whether DRG2 in the dopaminergic neurons is essential for motor coordination.

Minor Response: We have carefully proofread the manuscript and change the words by revised manuscripts.

“…, nucleus accumbens (NAc)” at page 8. Line 17 “The dot area indicates the quantification regions.” at page 8. Line 17-18, “(F), (G), (H)” at page 8. “…NAc … (Fig. 5D-E)” at page 9. Line 8 “(… 5F)” at page 9. Line 16, “(… 5G)” at page 9. Line 21,

Reviewer 2 Report

In the manuscript “DRG2 deficient mice exhibit impaired motor behaviors 2 with reduced striatal dopamine release” authored by Lim et al., studies were performed with regard to expression and function of DRG2 in the mouse. The study is well designed and the manuscript is well written, results are mostly well presented, and in the discussion the subject is comprehensively treated. However, there are some shortcomings which have to be clarified.

Fig. 2D, the lettering in one photomicrograph is HPF it should be Hip.

Fig. 3A, which region of the hippocampus is shown here?

S Fig. 1B, three different sections of the cerebral cortex reveal different intensities of DRG2 staining, has to be explained.

Fig. 5D, quantification of TH neuron number has to be explained. In addition, S Fig. 2, quantification of immunoreactivity has to be explained. 

Author Response

Response to Reviewers' 2 comments

Reviewer #2

Comment 1: Fig. 2D, the lettering in one photomicrograph is HPF it should be Hip.

Response 1: We very much appreciate the reviewer’s comment. We changed the “HPF” to “HIP” in the Fig. 2D lower panel.

Comment 2: Fig. 3A, which region of the hippocampus is shown here?

Response 2: The images in right panel of Fig. 3A were dentate gyrus (DG) region of hippocampus. We added this information to the figure legend.

Comment 3: S Fig. 1B, three different sections of the cerebral cortex reveal different intensities of DRG2 staining, has to be explained.

Response 3: We thank the reviewer for raising this point. To meet the comment raised by the reviewer, we conducted new experiments to get images with same exposure time and intensities of DRG2. We replaced the old images in Supplementary Fig. 1B with new ones.

Comment 4: Fig. 5D, quantification of TH neuron number has to be explained. In addition, S Fig. 2, quantification of immunoreactivity has to be explained. 

Response 4: According to your comment, we quantified the TH intensities and TH-positive neuron number of Fig. 5D and generated graphs based on the TH staining intensity and number of TH-positive neuron. We added these graphs to the Fig 5 as Fig 5E. We also added how to quantify TH intensity and TH-positive neuron to “Materials and Methods” section.

In addition, we add the sentence “TH neurons were stained with anti-TH antibody and detected using secondary antibody conjugated with horseradish peroxidase complex (Fig. 5D and 5E) or Alexa Fluor 488/594 (Supplemental Fig. 2C). The TH intensity and TH-positive neurons were quantified.” to the Results” section.

Figure 5 legends, page 8, line 18-20: “(E) Quantification of TH neurons in the images of (D). Graphs data represents relative TH staining intensity in the Str and NAc and number of TH positive neuron in the SN and VTA. Student’s t-test; ns, not significant.”

Minor Response: We have carefully proofread the manuscript and change the words by revised manuscripts.

“…, nucleus accumbens (NAc)” at page 8. Line 17 “The dot area indicates the quantification regions.” at page 8. Line 17-18, “(F), (G), (H)” at page 8. “…NAc … (Fig. 5D-E)” at page 9. Line 8 “(… 5F)” at page 9. Line 16, “(… 5G)” at page 9. Line 21,

Round 2

Reviewer 1 Report

I confirmed that the manuscript is adequately improved.

Reviewer 2 Report

The points raised by the reviewer are treated adequately. 
